# XVerse: Consistent Multi-Subject Control of Identity and Semantic Attributes via DiT Modulation

**Bowen Chen**,* **Mengyi Zhao**,* **Haomiao Sun**,* **Li Chen**,*,† **Xu Wang**,
**Kang Du**, **Xinglong Wu**

Intelligent Creation Team, ByteDance

{chenbowen.cbw, zhaomengyi.pl, sunhaomiao, chenli.phd, wangxu.ailab,
dukang.daniel, wuxinglong}@bytedance.com

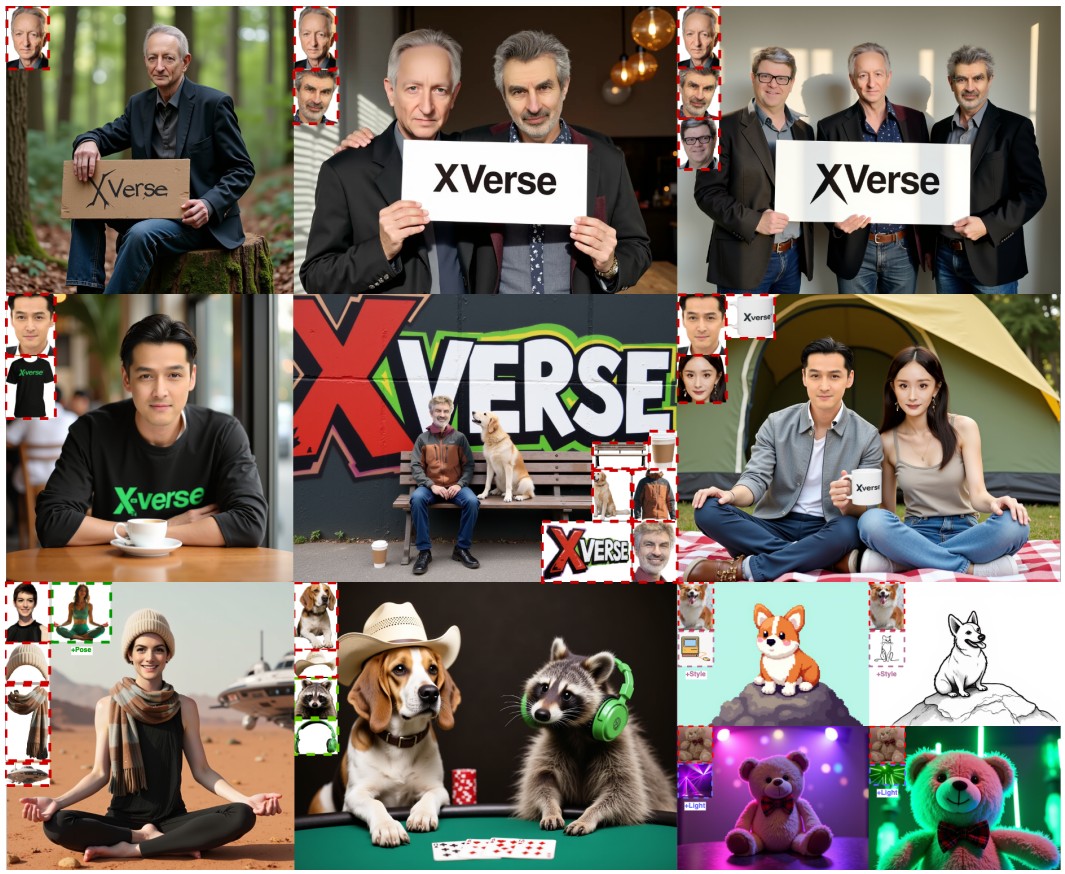

Figure 1: XVerse enables single/multi-subject personalization and the additional control of semantic attributes such as pose, style, and lighting. Input conditions are highlighted with red dots.

## Abstract

Achieving fine-grained control over subject identity and semantic attributes (pose, style, lighting) in text-to-image generation, particularly for multiple subjects, often undermines the editability and coherence of Diffusion Transformers (DiTs).

---

*Equal contribution.

†Corresponding author, project lead.

39th Conference on Neural Information Processing Systems (NeurIPS 2025).

Many approaches introduce artifacts or suffer from attribute entanglement. To overcome these challenges, we propose a novel multi-subject controlled generation model XVerse. By transforming reference images into offsets for token-specific text-stream modulation, XVerse allows for precise and independent control for specific subject without disrupting image latents or features. Consequently, XVerse offers high-fidelity, editable multi-subject image synthesis with robust control over individual subject characteristics and semantic attributes. This advancement significantly improves personalized and complex scene generation capabilities. Project Page: `https://bytedance.github.io/XVerse`; Github Link: `https://github.com/bytedance/XVerse`.

# 1 Introduction

The field of text-to-image generation has advanced remarkably [1; 2; 3; 4; 5; 6], enabling the creation of highly realistic and diverse images from textual descriptions. Initial breakthroughs in personalization focused on single subjects [7; 8; 9; 10]. These methods demonstrated strong control over individual subject appearances and showed good editability. However, the growing demand for more complex visual narratives and personalized content has spurred interest in extending these capabilities to scenarios involving multiple subjects within a single, coherent scene. This transition to multi-subject personalization presents substantial challenges, particularly in preserving individual identity fidelity and alleviating attribute entanglement.

Many current state-of-the-art multi-subject methods [11; 12; 13; 14] tried to leverage the attention mechanism in Diffusion Transformers (DiTs) [15] for injecting information from reference images. But this direct injection or strong reliance on image features can substantially impact the generation quality of the base model. This often leads to artifacts, distortions, attribute entanglement, and can compromise the overall structural integrity and coherence of the generated image. These limitations highlight a critical gap, underscoring the need for novel techniques that offer fine-grained, independent control over multiple subjects while preserving image quality, editability.

To address these limitations, this paper introduces XVerse, a novel method for consistent multi-subject control of identity and semantic attributes. We identify that the inherent modulation vectors within DiT blocks [15], which are typically employed for general conditioning, represent an underexplored yet highly promising pathway to achieve nuanced, subject-specific control. Building upon this insight, XVerse pioneers an approach centered on learning offsets within the text-stream modulation mechanism of DiTs. With the reference images provided, XVerse utilizes an adapter to transform them into share offsets and per-block offsets for token-specific text-stream modulation. This technique allows for condition injection from diverse reference images while preserving the image's underlying structural integrity. To enhance fine-grained details, we incorporate VAE-encoded image features into the single-stream block of FLUX [6]. Instead of being the main conditioning factor, the VAE-derived features play a supporting role in enhancing details for the backbone network. This strategy successfully minimizes the occurrence of artifacts and distortions, enabling XVerse to achieve exceptional multi-subject controlled generation results (as shown in Figure 1). Extensive testing conducted on our benchmark validates XVerse's exceptional performance in terms of both flexibility of editing and maintaining the appearance of the subject.

Our contributions are summarized as follows:

- We propose XVerse, a novel framework for fine-grained multi-subject controllable generation. Our approach integrates the reference images into the text-stream modulation offsets. Additionally, we leverages VAE-encoded image features to enhance fine-grained details which are difficult to expresss in the semantic space. This allows XVerse to achieve a high degree of consistency with reference images, while preserving the original diffusion model's editability by maintaining the image composition.

- Through training on a high-quality datasets constructed by our data pipeline, XVerse achieves outstanding performance in generation tasks under multi-subject control. Moreover, due to the flexibility of text-stream modulation, XVerse also demonstrates strong generalization performance in tasks such as maintaining semantic information such as posture, lighting, and background.

- We present a comprehensive benchmark XVerseBench, which evaluates both single-subject and multi-subject controlled image generation. This benchmark provides a rigorous methodology for assessing a model's ability to support flexible editing, maintain subject characteristics, and preserve distinct identities.

## 2 Related Work

### 2.1 Subject-driven Generation

Subject-driven generation tasks aimed at synthesizing user-specific content, and have made significant progress in recent years. Early endeavors predominantly concentrated on single-subject personalization. These approaches can be broadly classified into two main categories: (1) Fine-tuning-based methods, such as Textual Inversion [7] and DreamBooth [8], which adapt pre-trained models to embed novel concepts from a few exemplar images of a single subject. (2) Tuning-free methods, including IP-Adapter [9] and Photoverse [10], leverage powerful vision encoders to inject subject identity directly into the generation process without test-time fine-tuning, thereby offering enhanced flexibility. However, extending personalization to accommodate multiple subjects within a single, cohesive image presents substantial challenges, particularly in preserving individual identity fidelity and mitigating attribute entanglement. Recent progress leveraging DiTs [15] has begun to address these complexities. For instance, OmniControl [11] demonstrated versatile control by conditioning DiT inputs. Concurrently, a line of research has focused on unified frameworks for complex multi-subject image customization, exemplified by UniReal [12], UNO [13], and DreamO [14]. While these methods significantly advance multi-subject generation, their primary strategies involve conditioning the input token sequence, imposing explicit constraints, or modifying attention mechanisms to guide the generative process. XVerse introduces a distinct perspective, operating as a tuning-free method for multi-subject personalization. With the utilization of the text-stream modulation mechanism in DiTs, XVerse can enable precise, subject-specific conditioning while preserving the structural integrity of the generated image

### 2.2 Modulation in Generative Models

Modulation mechanisms have been instrumental in enhancing the controllability and expressiveness of generative models. This concept involves dynamically adjusting a model's internal activations or parameters based on conditioning information. StyleGAN [16] pioneered the use of modulation by introducing adaptive instance normalization (AdaIN), which modulates features using a style vector. A key discovery associated with this approach was that even small modifications to these modulation parameters could induce smooth and semantically meaningful perturbations in the generated image. This insight spurred a wave of research focused on leveraging modulation layers, leading to significant advancements and successful applications in tasks such as image editing and manipulation [17; 18; 19; 20; 21].

Transformer-based models, such as DiTs, commonly employ modulation mechanisms like adaptive layer normalization (AdaLN) [22]. These mechanisms allow conditioning information (e.g., text embeddings, timesteps) to guide image generation by modulating normalization layers within transformer blocks. Tokenverse [23] explored modulation for subject preservation, they typically rely on fine-tuning strategies. Such methods necessitate extensive training on predefined datasets to learn subject-specific modulations, thus limiting their adaptability to novel, unseen subjects without retraining or inference-time optimization. In contrast, XVerse is designed to inject rich identity and semantic information for arbitrary subjects without such subject-specific fine-tuning.

## 3 Method

### 3.1 Preliminaries

DiTs have introduced significant advancements in the quality, efficiency, and scalability of image synthesis, establishing DiTs as the foundational architecture for most state-of-the-art models. The Attention Block in DiTs process text and image tokens concurrently within their transformer layers. This attention mechanism provides a pathway for injecting control signals, such as features from control images, into the token representations. However, while injecting control signals through

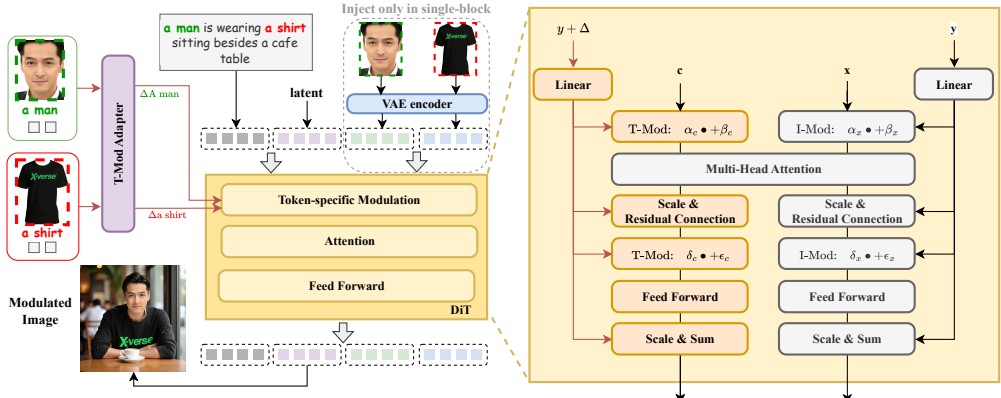

Figure 2: Overview of the XVerse framework. The reference images are processed by a T-Mod Resampler and subsequently injected into the per-token modulation adapter. Additionally, to supplement image details, the VAE-encoded features of the reference image are also utilized as input to the single-stream block of DiTs.

attention can achieve good similarity improvements, this approach can also cause the model's sampling trajectory to deviate, leading to a reduction in image generation quality. In this work, We try to further explore the potential of the modulation mechanism in DiTs to achieve more precise subject control editing.

**Modulation Mechanism in DiTs.** The modulation mechanism in DiTs, adapted from techniques popularized by StyleGAN[16], refines neural network activations by applying learned scale factors and bias terms. For instance, models like Stable Diffusion 3 [15] and FLUX [6] utilize a Multi-Layer Perceptron (MLP) to process both the diffusion timestep $t$ and a CLIP [24] embedding of the text prompt (e.g., $f_p = \text{CLIP}(p)$), thereby deriving a conditioning vector $y$:

$$y = \text{MLP}(t, f_p). \tag{1}$$

This vector $y$ is then further processed, typically through a linear layer, to generate a set of modulation parameters. In the DiT architecture, this yields twelve distinct parameters for each block. Six of these parameters modulate text features, and the remaining six modulate image features. The integration of these modulations is commonly achieved using techniques such as Adaptive Layer Normalization (AdaLN) and residual connections. AdaLN is implemented prior to the Attention Layer and Feed-Forward Layer, and can be expressed as:

$$\text{AdaLN}(x, \alpha, \beta) = \alpha \cdot \frac{x - \mu(x)}{\sigma(x)} + \beta. \tag{2}$$

where $x$ is the input feature, $\mu(x)$ and $\sigma(x)$ are its mean and standard deviation respectively, and $\alpha$ (scale) and $\beta$ (bias) are modulation parameters derived from the conditioning vector $y$. Residual connections are added after the Attention layer and the Feed-Forward layer. They help stabilize training and enable deeper networks, typically taking the form:

$$x_{out} = \gamma \mathcal{F}(x_{in}) + x_{in}. \tag{3}$$

where $x_{in}$ is the input to a layer or block, $\mathcal{F}(x_{in})$ is the learned residual mapping, $x_{out}$ is the output, and $\gamma$ is a scaling parameter applied to the residual term, also derived from the conditioning vector $y$. These mechanisms facilitate fine-grained control over feature representations at various stages within the network. Such adaptive modulation significantly enhances the model's capacity to align image generation with the conditioning inputs.

An essential aspect of this modulation mechanism is its ability to operate separately from the main data flow of the attention mechanism. This separation allows for the precise integration of visual feature representation into specific words without interfering with the denoising process. This has the potential to minimize errors and enhance the overall quality of the generated image. Furthermore, text-based features often offer clearer semantic directionality compared to image-based features. This indicates that manipulating text-stream modulation signals can offer a more straightforward and comprehensible method to control the generative process. Consequently, XVerse focuses on leveraging text-side feature modulation to attain precise control over image synthesis.

## 3.2 Multi-subject Controled Generation

The framework of XVerse is shown in Figure 2. We first leverage the modulation mechanism for its robust preservation of key attributes from the reference images. Subsequently, we inject fine-grained details with the VAE-encoded image features in single-stream blocks of FLUX model. The subsequent sections will detail these enhancements.

**Enhanced Text-Stream Modulation with Image Feature Control.** To enhance fine-grained control over the generation process, we augment the existing text-stream modulation framework by introducing image features as an additional control signal. Specifically, given a conditioning image $I_c$, we first extract its deep features using the CLIP model, represented as $f_c = \text{CLIP}(I_c)$. Subsequently, we combine these image features with the CLIP-encoded features $f_p$ of the text prompt, using a perceiver resampler [25] as the text-stream modulation adapter (T-Mod Adapter). Here, the text features $f_p$ serve as the query vector, and the resampler outputs an offset $\Delta_{\text{cross}}$:

$$\Delta_{\text{cross}} = \text{Resampler}(f_p, f_c). \tag{4}$$

This output, $\Delta_{\text{cross}}$, encapsulates synergistic information from both the textual prompt and the conditioning image, acting as a corrective or refining signal. For example, if the text prompt $p$ is about a "handbag" and the accompanying image $I_c$ shows a "brown leather handbag", $\Delta_{\text{cross}}$ helps the model adjust its representation from a generic handbag to the specific "brown leather handbag" with a series of visual characteristics (such as material, color, and style).

Crucially, $\Delta_{\text{cross}}$ is formulated as an offset vector targeting specific tokens. This offset is added to the corresponding token embeddings injected into the model, enabling precise control over the semantics of each token while preserving the structure of the text-to-image result. The adjusted conditioning signal $y^*$ is thereby derived:

$$y^* = \text{MLP}(t, f_p) + \Delta_{\text{cross}}. \tag{5}$$

This adjusted conditioning signal $y^*$ is then utilized to adjust the original modulation parameters (e.g., scale and shift parameters) applied to the network's activations, thereby refining the conditioning influence. Since our method primarily performs injection in the textual space, it can naturally generalize to the control of high-level semantic attributes such as pose, lighting, and style, without requiring additional differentiation for these conditions.

To further enhance the level of control over the final image output, we draw inspiration from StyleGAN's expansion of the $\mathcal{W}$ latent space to $\mathcal{W}^+$, enabling each Transformer block $i$ in our model to receive customized conditioning. This is achieved by decomposing the $\Delta_{\text{cross}}$ signal. Instead of a single offset, we compute a shared component, $\Delta_{\text{shared}}$ (applied across all DiT blocks), and individual components for each block $i$, denoted as $\Delta^i_{\text{per-block}}$ (which can be computed on a per-block or per-stage basis):

$$y^*_i = \text{MLP}(t, f_p) + \Delta_{\text{shared}} + \Delta^i_{\text{per-block}}. \tag{6}$$

This structured decomposition of the image-derived text offset facilitates more precise and adaptive control over the influence of text conditioning at various levels of the generation process.

**Refined Attention Module with Controlled VAE Integration.** While using only text-stream modulation can achieve good editability and generation results, its ability to preserve detailed information is still limited. Inspired by OmniControl [11], we introduce VAE-encoded image features as an auxiliary module to further enhance the capability of our approach for maintaining consistency of fine-grained features. To avoid potential negative impacts of directly injecting image features (such as artifacts or degradation of image quality), we constrain the role of VAE features, making them primarily an auxiliary module for supplementing image details, rather than the dominant mechanism for feature injection. Specifically, we restrict the injection of VAE features only to single-stream blocks within the FLUX model. To effectively distinguish different image patch regions within the latent space of the conditioning image, the position index for the latent is changed from $(i, j)$ into individual index as UNO[13].

## 3.3 Regularizations

In modulation space $\mathcal{M}^+$, feature vectors derived from highly similar subjects often become entangled, leading to subject confusion and unintended fusion in generated results. To address this, we

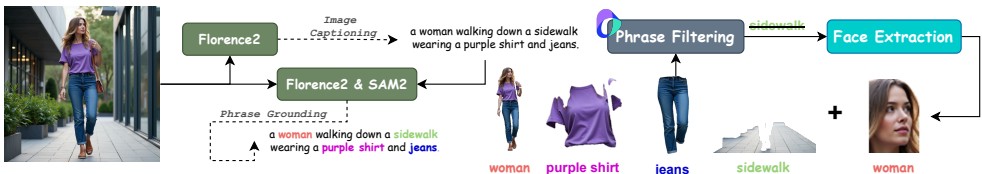

Figure 3: Training Data Construction Pipeline.

introduce two critical regularization techniques: region preservation loss and text-image attention loss, which facilitate multi-subject disentanglement.

**Region Preservation Loss.** We construct a new training sample by randomly selecting two existing samples and concatenating them in a side-by-side left-right configuration, with their captions combined into a unified overall caption. We randomly retain modulation injection from only one side (left or right), rather than both. For regions without modulation injection, we enforce consistency between the model's output and the T2I branch's output in those regions using L2 loss.

$$\mathcal{L}_{\text{region}} = \mathbb{E}_{x,y} \left[ ||(1 - M_c) \odot (V_{\theta'}(z_t, t, y^*) - V_\theta(z_t, t, y))||_2^2 \right] \tag{7}$$

where $z_t$ denotes the noisy latent at timestep $t$, $V_\theta(z_t, t, y)$ denotes T2I branch's output, $V_{\theta'}(z_t, t, y^*)$ denotes the XVerse's output, and $1 - M_c$ denotes the non-modulated mask region. This approach not only regularizes subject-specific features but also serves as a data augmentation strategy for multi-subject datasets, enhancing the model's ability to distinguish and preserve subject characteristics.

**Text-Image Attention Loss.** To maintain the compositional and editability properties of the T2I branch after modulation injection, we align the cross-attention dynamics between two branches. Specifically, we compute an L2 loss with normalization over the text-image cross-attention maps of the modulated model and the reference T2I branch. This encourages the modulated model to retain attention patterns that closely match those of the T2I branch, ensuring that semantic interactions between text and image regions remain consistent and editable.

## 3.4 Collection of Training data

High-quality multi-entity data remains scarce, and modulation injection necessitates knowledge of the corresponding text tokens in the target prompt for each conditional image. To address this, we introduce a high-quality general-purpose multi-entity data annotation pipeline.

**Single-Image Data Construction.** We curated a 1M-scale dataset of images with resolutions exceeding 512 pixels from LAION [26], constructing a universal multi-entity single-image dataset via the data workflow depicted in 3. Specifically, we first employ Florence2 [27] for joint image caption generation and phrase grounding, followed by large language model (LLM)-driven phrase filtering and classification to exclude non-entity terms (e.g., "sky", "water surface") that do not align with our definitional criteria. Subject segmentation is then performed using SAM2 [28], with additional face detection and extraction applied specifically for human entities. While open-source data includes diverse scenarios, challenges such as suboptimal aesthetic quality and scene complexity impede model learning. To mitigate this, we supplement the dataset with an additional 1M-scale corpus of high-aesthetic-quality images synthesized using FLUX [6].

**Cross-Image Data Construction.** For human-centric data, we harvested a 100K-scale single-person multi-view dataset from proprietary in-house collections, forming up to three image pairs per subject ID. For general objects, we leveraged the Subject200K [11] dataset. The construction workflow mirrors that of single-image data, with domain-specific refinements: human data incorporates ID similarity thresholds, while general object data employs DINOv2 [29] similarity filtering—both applied independently to enforce cross-view consistency.

# 4 Experiments

## 4.1 Experiments Setting

**Implementation Details.** In XVerse, we design a three-stage training pipeline to achieve precise multi-subject controlled image generation. We first train the text-stream modulation adapter to

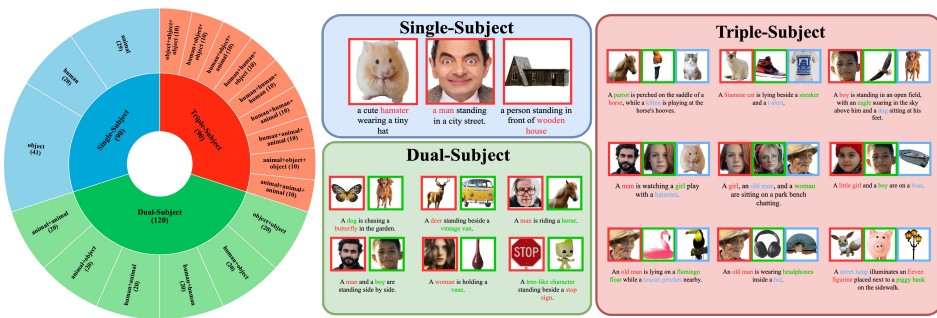

Figure 4: Data distribution and samples for XVerseBench. XVerseBench includes evaluations of single-subject, dual-subject, and triple-subject controlled image generation. The figure also illustrates the number of test samples allocated to each category.

establish foundational semantic alignment. It serves as the basis for conditional feature injection which ensures high-level consistency between injected images and generated results. Building upon the first stage, we introduce VAE-encoded features to enhance detail preservation. While maintaining the global structure and key characteristics obtained from stage one, this phase focuses on injecting fine-grained visual details through hierarchical feature fusion. The first two stages employ hybrid training data containing both single-subject and multi-subject samples from our single-image dataset. However, we found that such a reconstruction paradigm tends to produce copy-paste effects during inference, resulting in reduced diversity and compromised structural fidelity. To address this issue, we propose a third training phase incorporating cross-image data where the subject appearances differ between injected and target images. This extension enhances the capability of our approach to establish robust appearance mapping under heterogeneous visual conditions. Throughout all stages, we maintain mixed training on both single-subject and multi-subject datasets.

Based on FLUX.1-dev [6] text-to-image generation model, we employ LoRA [30] with a rank of 128 to efficiently fine-tune the model while maintaining its generalization capabilities. We utilize two three-layer resamplers with an intermediate dimension of 3072 to generate the shared offsets and per-block offsets for text-stream modulation. The model was trained for 70K, 150K, and 10K iterations in the respective stages. Both the text-stream modulation adapter and LoRA layers were optimized with a learning rate of 5e-6. The region preservation loss was assigned a weight of 10, while the text-image attention loss was weighted at 0.01. Training was conducted on 64 NVIDIA A800 GPUs (40GB each), taking approximately 7 days in total — distributed as 2 days for Stage 1, 4 days for Stage 2, and 1 day for Stage 3.

**XVerseBench Details.** Existing controlled image generation benchmarks often focus on either maintaining identity or object appearance consistency, rarely encompassing datasets that rigorously test both aspects. To comprehensively assess the models' single-subject and multi-subject conditional generation and editing capabilities, we constructed a new benchmark by merging and curating data from DreamBench++ [31] and Unsplash50 [32].

Our resulting benchmark XVerseBench comprises 20 distinct human identities, 65 unique objects, and 45 different animal species/individuals. To thoroughly evaluate model effectiveness in subject-driven generation tasks, we developed test sets specifically for single-subject, dual-subject, and triple-subject control scenarios. This benchmark includes 300 unique test prompts covering diverse combinations of humans, objects, and animals. Figure 4 shows more detail information and samples for each categories. For evaluation, we employ a suite of metrics to quantify different aspects of generation quality and control fidelity: including DPG score [33] to assess the model's editing capability, Face ID similarity [34] and DINOv2 [29] similarity to assess the model's preservation of human identity and objects, and Aesthetic Score [35] to measure to evaluate the aesthetics of the generated image. XVerseBench aims to provide a more challenging and holistic evaluation framework for state-of-the-art multi-subject controllable text-to-image generation models.

## 4.2 Comparisons with State-of-the-art Methods

We compare our proposed XVerse method with several leading multi-subject driven generation techniques, including MS-Diffusion [36], MIP-Adapter [37], OmniGen [38], UNO [13], and DreamO

Table 1: Quantitative results of single-subject and multi-subject driven generation on XVerseBench.

| Method | Single-Subject | | | | | Multi-Subject | | | | | Overall |
|---|---|---|---|---|---|---|---|---|---|---|---|
| | DPG | ID-Sim | IP-Sim | AES | AVG | DPG | ID-Sim | IP-Sim | AES | AVG | |
| MS-Diffusion [36] | 96.94 | 6.58 | 51.06 | **59.69** | 53.57 | 87.27 | 4.81 | 40.90 | **55.87** | 47.21 | 50.39 |
| MIP-Adapter [37] | 77.48 | 28.39 | 66.32 | 52.09 | 56.07 | 84.52 | 19.49 | 49.89 | 51.78 | 51.42 | 53.75 |
| OmniGen [38] | 85.19 | 60.17 | 70.73 | 51.89 | 67.00 | 81.71 | 42.18 | 52.11 | 51.35 | 56.84 | 61.92 |
| UNO [13] | 91.82 | 37.22 | **74.35** | 55.21 | 64.65 | 87.57 | 26.00 | 60.62 | 53.04 | 56.81 | 60.73 |
| DreamO [14] | **97.51** | 58.74 | 67.69 | 53.80 | 69.44 | 89.75 | 44.21 | 60.87 | 51.16 | 61.50 | 65.47 |
| **XVerse (Ours)** | 93.50 | **63.02** | 71.35 | 56.63 | **71.13** | 91.77 | 51.03 | 61.04 | 53.68 | 64.38 | **67.76** |

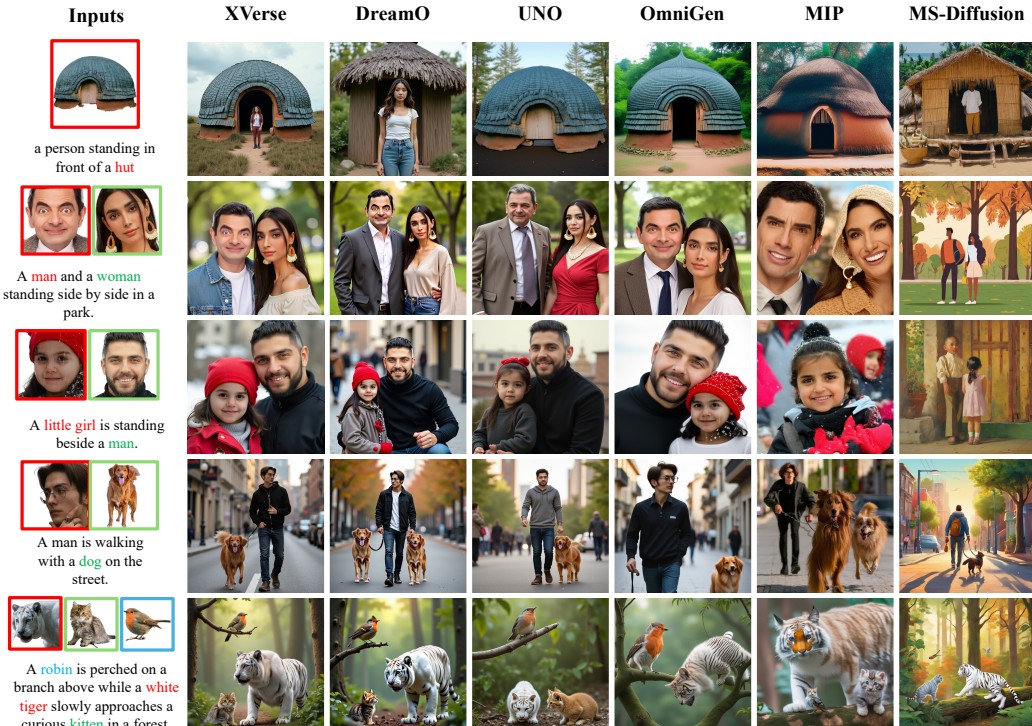

Figure 5: Qualitative comparison with different methods on XVerseBench.

[14]. The generation results are evaluated for both single-subject and multi-subject tasks, with quantitative findings presented in Table 1. Our XVerse method achieves the highest Overall score of 67.76 , significantly outperforming all other compared methods. This clearly indicates a strong comprehensive advantage of our approach.

In the single-subject generation category, XVerse demonstrates exceptional performance, securing the top AVG score of 71.13 . This underscores its robust capability in generating high-quality images focused on individual subjects. Notably, XVerse achieves the best identity similarity score (ID-Sim) of 63.02, suggesting superior preservation of subject identity. While DreamO leads in DPG with 97.51, XVerse's strong average performance, bolstered by competitive scores in IP-Sim (Object Similarity) at 71.35 and AES (Aesthetic Score) at 56.63, highlights its well-rounded excellence.

Meanwhile, XVerse truly excels in the more challenging multi-subject generation tasks, achieving a leading AVG score of 64.38. This remarkable performance can be attributed to XVerse's novel approach of learning offsets within the text-stream modulation mechanism of DiT. This allows for precise conditioning from diverse image types while crucially preserving the image's structural integrity. Furthermore, the careful integration of VAE-derived features for detail refinement, rather than dominant conditioning, effectively mitigates artifacts and distortions, which is particularly vital for maintaining clarity and attribute disentanglement in multi-subject scenarios.

Figure 5 presents a qualitative comparison of our method with other state-of-the-art approaches. As illustrated, our model demonstrates a superior capability in maintaining the consistency and relevance

Table 2: Ablation Study of Joint Training of Text Modulation and VAE-Encoded Features

| EXP ID | EXP Name | Init | Steps | Single-AVG | Multi-AVG | AVG |
|--------|----------|------|-------|------------|-----------|-----|
| 1 | Text Modulation Only | Scratch | 30K | 57.55 | 52.73 | 55.14 |
| | | | 40K | 58.30 | 53.54 | 55.92 |
| 2 | Single-Stream VAE Only | Scratch | 30K | 60.71 | 53.32 | 57.02 |
| | | | 40K | 59.60 | 53.37 | 56.49 |
| 3 | Text Modulation + VAE | Exp 1, 30K | 10K | 62.62 | 55.16 | 58.89 |

Table 3: Ablation Study of Different Regulation Losses

| EXP | Steps | Single-AVG | Multi-AVG | AVG |
|-----|-------|------------|-----------|-----|
| Baseline | 30K | 57.55 | 52.73 | 55.14 |
| Baseline w/o RPL | 30K | 55.47 | 50.72 | 53.10 |
| Baseline w/o TIAL | 30K | 53.42 | 44.51 | 48.97 |

between identities and associated objects within the generated images. This is a direct result of our refined modulation strategy. By carefully adjusting the modulation offsets, XVerse achieves enhanced text-image alignment, which is particularly evident in its accurate depiction of object quantities and the relationships between multiple subjects. Furthermore, when compared to existing methods, our model consistently produces images with a higher degree of naturalness and visual plausibility. This improved realism underscores the advantages of our approach in editing the text-stream modulation pathway, allowing for more faithful and aesthetically pleasing image synthesis.

## 4.3 Ablation Studies

To further evaluate the effectiveness of different modules in XVerse and analyze the impact of key components, we conduct a series of ablation studies. All quantitative comparisons are performed under the same configuration (e.g., hyperparameters, training schedule), unless otherwise specified. Due to computational resource limitations, all experiments are executed on 16 NVIDIA A800 (40GB) GPUs.

**Joint Training of Text Modulation and VAE-Encoded Features** We test three setups to examine text-VAE collaboration: (1) Text Modulation Only, (2) Single-Stream VAE Only, (3) Joint Training. The results are shown in Table 2. After 30K steps, Text Modulation Only scores 57.55 (single-subject) and 52.73 (multi-subject) AVG; extending to 40K steps (additional 10K) brings marginal gains (58.30/53.54). Single-Stream VAE Only outperforms text-only at 30K (60.71/53.32) but degrades at 40K, reflecting standalone VAE instability (attribute blending disrupts consistency). Notably, Experiment 3 (Joint Training, initialized from Exp.1's 30K checkpoint, 10K extra steps) boosts scores to 62.62 (single) and 55.16 (multi), mitigating VAE injection instability and enabling more disentangled generation. Figure 6 shows the qualitative comparison in both dual-subject and triple-subject controlled generation tasks. The experimental results show that by converting the reference image to text-stream modulation offset vectors, XVerse achieves the personalization of the condition subjects while maintaining the composition identical to the original T2I outcomes. Additionally, the VAE features play a supportive role by helping the model add specific details without compromising the overall structure and semantic coherence established by the text-stream modulation. This collaborative approach allows XVerse to maintain subject consistency while achieving a high degree of editability.

**Impacts of Different Regularization Losses.** To verify the necessity of Region Preservation Loss (RPL) and Text-Image Attention Loss (TIAL), we compare the baseline (both losses) with RPL-ablated and TIAL-ablated models. Other configurations (e.g., training steps) are identical to isolate loss impacts. As shown in Table 3, the baseline achieves the highest AVG (single: 57.55, multi: 52.73). RPL ablation drops performance by 2.04% (as it helps to maintain category consistency), while TIAL ablation causes a 7.17% decline (as it ensures precise attribute localization, reducing irrelevant feature leakage). This confirms RPL and TIAL are complementary and indispensable.

Table 4: Ablation Study of Different VAE-encoded Feature Injection Strategies

| EXP | Steps | Single-AVG | Multi-AVG | AVG |
|---|---|---|---|---|
| Single-stream, VAE Only | 30K | 60.71 | 53.32 | 57.02 |
| All Blocks, VAE Only | 30K | 58.58 | 51.35 | 54.97 |

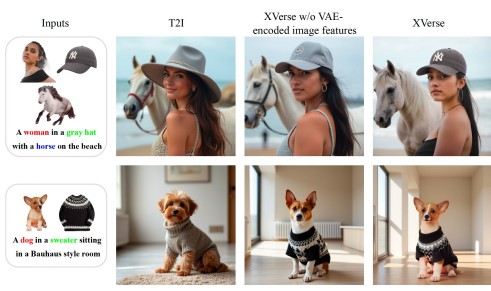

Figure 6: Effect of text-stream modulation resampler and VAE-encoded image features.

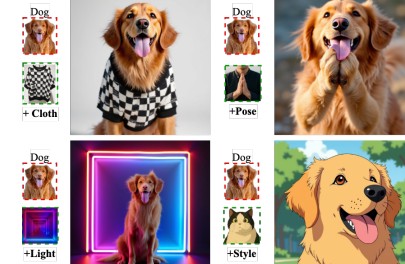

Figure 7: The control of semantic attributes, such as cloth, pose, lighting, and style.

**VAE-encoded feature injection blocks.** We evaluate two VAE injection strategies to identify the optimal configuration: "Single-stream, VAE Only" (injecting exclusively into single-stream blocks) and "All Blocks, VAE Only" (injecting into both double-stream and single-stream blocks). As shown in Table 4, the single-stream injection consistently outperform the all-blocks approach, achieving average scores of 60.71 versus 58.58 for single-subject evaluations, and 53.32 versus 51.35 for multi-subject tests. The inferior performance of the all-blocks injection is attributed to feature interference, where redundant signals disrupt the core generation process. Focusing injection on the core single-stream pathways results in better alignment and overall effectiveness compared to a more diffuse injection strategy.

### 4.4 Applications

Figure 7 demonstrates XVerse's capability to control semantic attributes of the generated image. XVerse exhibits precise control over attributes such as lighting, subject posture, clothing, and artistic style. By injecting reference images toward targeted words, XVerse can manipulate these semantic attributes without the need for extensive training data on specific attribute categories. These results further underscore the model's exceptional ability to generalize and edit effectively.

## 5 Conclusion

In this paper, we introduce XVerse, an innovative framework designed to excel in the complex task of precise multi-subject control within DiTs. XVerse injects reference image features through modulation offsets and control the token-specific representation within the DiT blocks. This approach facilitates adaptive, per-subject conditioning by precisely governing how textual embeddings are transformed and integrated throughout the diffusion process. Consequently, XVerse effectively mitigates common generation issues such as attribute entanglement and artifacts, demonstrating particular excellence in both single-subject and multi-subject controlled generation tasks.

**Limitations.** While XVerse showcases significant progress in fine-grained multi-subject control, there are certain limitations that need to be addressed and opportunities for future research. One major challenge is the lack of high-quality, large-scale cross-image multi-subject datasets, which are essential for training and evaluating models that can understand and generate complex interactions between subjects. Additionally, our current research has mainly focused on the text-stream modulation pathway, leaving room for exploration in utilizing image-modulation techniques for precise pixel-level or region-specific control.

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

# A    Samples for Training Dataset

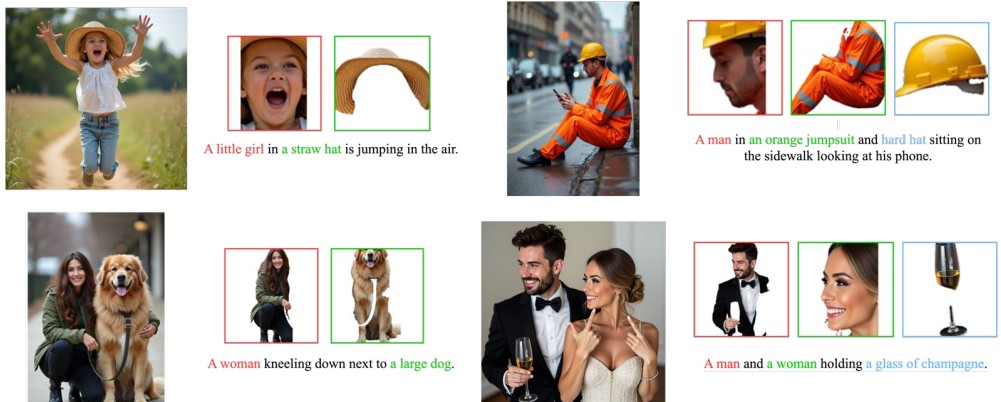

Figure 8: Examples of training data for multi-subject controlled generation.

Figure 8 presents examples of our training data for multi-subject controlled generation. As illustrated, the dataset covers a diverse range of scenarios, including human-object interactions, human-animal compositions, and complex multi-person scenes. For human-centric data, we intentionally randomly select facial images or full-body images as reference inputs. This strategy can further enhance the model's generalization performance. By utilizing this diverse and extensive dataset, which encompasses a wide range of scene variations and control types, our model is able to achieve impressive editing capabilities while maintaining high consistency with the reference images.

# B    Impact of Prompt Variation on the Generated Image

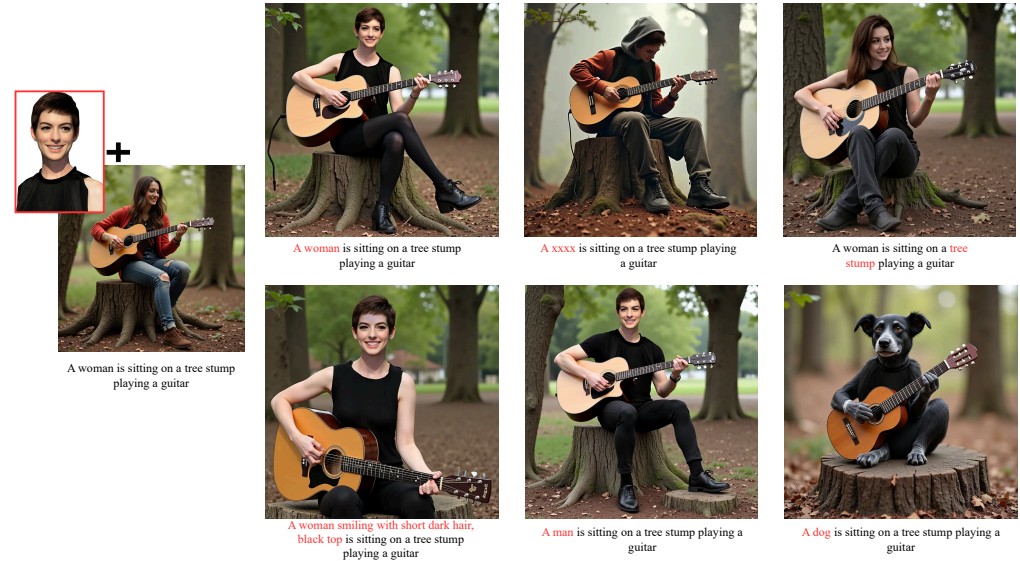

Figure 9: Impact of prompt variation on subject-controlled Image generation. The reference image and the initial output of our text-to-image generation model are shown on the left side. The right side illustrates the influence of different prompts on the generated output, with the prompt variances highlighted in red.

To evaluate the impact of prompt variation on subject-controlled image generation, we modified the injected words in the per-token text-modulation module while keeping the reference image constant. The results, shown in Figure 9, offer valuable insights. This visualization effectively illustrates

that a more detailed prompt description improves the preservation of the subject's identity in the generated image. Additionally, when the prompt closely matches the reference image, our model not only incorporates intricate image details but also maintains a high level of control over the subject's attributes, even allowing for successful changes such as gender. On the other hand, if there is a significant semantic mismatch between the injected prompt and the reference image (e.g., trying to generate a person's image from prompts like "a dog" or "a tree stump"), the injection process consistently fails. This highlights our model's ability to accurately target and incorporate reference image features into specific words, enabling precise control over the generated output.

## C   Comparison of the CLIP-T and DPG scores

When evaluating text-to-image generation models, the CLIP-T [24] score has been a prevalent metric in prior studies, assessing semantic consistency by leveraging CLIP's image-text embeddings. However, our research highlights the superior efficacy of the DPG (Dense Prompt Graph) score, particularly for intricate prompts. While CLIP-T offers a broad measure of semantic alignment, the DPG score is specifically designed to evaluate a model's capacity to interpret and execute detailed and complex textual instructions. It rigorously assesses editing abilities across multiple objects, diverse attributes, and intricate relationships, thereby capturing the nuanced and fine-grained semantic alignment crucial for advanced compositional generation. This provides a more comprehensive and robust evaluation for challenging scenarios.

## D   Illustration of Region Preservation Loss

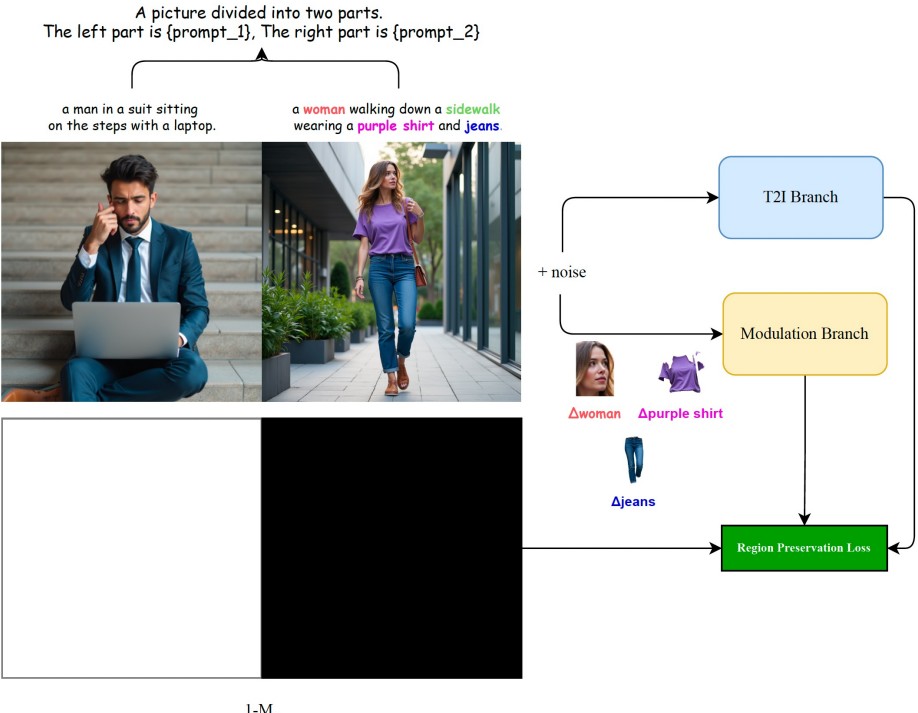

Figure 10: Illustration of the region preservation loss.

Figure 10 shows the illustration of our region preservation loss. We form training samples by concatenating two existing samples, merging their captions, and randomly applying modulation to only one side. For the unmodulated regions, defined by $M_c$, we enforce consistency between our model's output ($V_{\theta'}(z_t, t, y^*)$) and the text-to-image branch's output ($V_\theta(z_t, t, y)$) via an L2 loss (Eq. 1). By using this regularization, XVerse can better inject the reference image into specific areas without affecting the generation of irrelevant areas, thereby achieving more precise generation control.

# E    Ablation Study for Text-Image Attention Loss

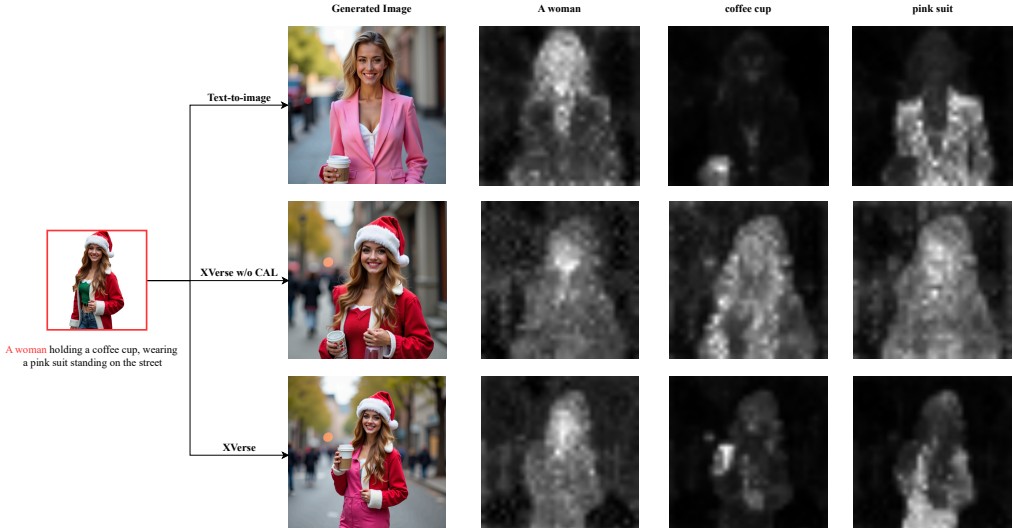

Figure 11: The qualitative comparsion of Text-Image Attention Loss. This image shows the generated results and attention maps for "woman", "coffee cup", and "pink suit" for each method.

To validate the effectiveness of our text-image attention loss, we conducted an experiment where we excluded this regularization and examined the generated outputs along with their respective attention maps. The qualitative analysis presented in Figure 11 clearly demonstrates the significance of this method. It demonstrates our method's ability to maintain the structural and editable characteristics of the T2I branch following modulation injection. Through ensuring L2 consistency between the cross-attention maps of the modulated model and the reference T2I branch, our approach ensures the reliable preservation of text-image semantic interactions. This ultimately enables precise control over semantics, as visually evidenced by the generated results and attention maps for specific prompts like "woman," "coffee cup," and "pink suit."

# F    Broader Impacts

Our model, XVerse, marks a significant leap in multi-subject controllable text-to-image generation, leading to enhanced fidelity and editability. This breakthrough holds substantial positive societal impacts, particularly within the creative industries, where it can revolutionize the creation of personalized and complex visual content. Furthermore, XVerse can transform education and training by providing more engaging and tailored visual aids, and contribute to content inclusivity by enabling the representation of a wider range of individuals and scenarios.

However, this powerful technology also presents potential negative societal impacts. The improved generation capability could lead to misinformation and deepfakes, raise privacy concerns if used improperly, and potentially amplify biases present in training data. As foundational research, XVerse isn't directly tied to deployment. Yet, we believe it's crucial to acknowledge these risks. Future work will explore mitigation strategies like content detection and ethical guidelines, contributing to the responsible advancement of generative AI.

