# OpenReview forum: "XVerse: Consistent Multi-Subject Control of Identity and Semantic Attributes via DiT Modulation"
_NeurIPS.cc/2025/Conference — NeurIPS 2025 poster_

### Official Review · Reviewer_yqTe · 2025-06-14

**Clarity:** 3
**Significance:** 2
**Originality:** 2
**Rating:** 4
**Confidence:** 5

**Summary:**

This paper investigates the subtask of text-to-image generation—target customization—by projecting target identities into the text modulation space of Flux to achieve this goal.

**Questions:**

See Weaknesses

**Ethical Concerns:**

["NO or VERY MINOR ethics concerns only"]

**Final Justification:**

Having carefully studied the author's rebuttal, their response has largely dispelled my doubts. The author frankly explained the "Copy-and-Paste" effect and appropriately discussed the importance of network architecture and data. I am well aware of the difficulties involved here, and on the whole, this paper has achieved a good level of completion. Therefore, I have finally decided to change the result to "borderline acceptance".

**Limitations:**

yes

**Quality:**

3

**Strengths And Weaknesses:**

Advantages:
1. The paper is fluently written with clear expressions, and the figures are appealing.

Disadvantages:
1. My primary concern is that projecting target identities into Flux's text modulation space, in my view, is fundamentally identical in principle to the numerous previous methods that projected target identities into the original text representation space. The only difference is that the modulation space here is a secondary projection of the text space. This explains why, similar to past approaches, projecting target identities solely into the text space results in insufficient fidelity, necessitating the concatenation of image VAE features.
2. Regarding the ablation study, I am more interested in the differences between projecting target identities into Flux's original text representation space and its text modulation space.
3. Using VAE-encoded image features to enhance target identity fidelity has become a widely adopted technique.
4. The generated images in this paper exhibit a pronounced "copying" effect. As shown in Figure 1, the characters' postures and expressions are highly similar to the reference images.
5. Here's a question: In the current task of target-customized characters, do you think network architecture or data is more important? I speculate that a primary reason for the obvious copying effect in this paper is the insufficient diversity of target samples during data construction. Although the third-stage training in this paper aims to alleviate this issue, the problem persists.

---

> ### Author Rebuttal · Authors · 2025-07-31
>
> Thank you for your comprehensive review and thoughtful questions. Your feedback is instrumental in enhancing the clarity and depth of our work. We have addressed each of your comments below:
>
> 1. **Differences from text representation space feature injection methods**
>
> 	Thank you for your insightful observation. While there may seem to be similarities in projecting identities into text-related spaces, our approach differs fundamentally from prior methods like PhotoMaker.
>
> 	Previous strategies often directly modify token values through addition or subtraction, which can disrupt the model's inherent text-to-image structure and compromise editability. In contrast, XVerse introduces perturbations to the distribution within the text modulation space, minimizing interference with the base model's structural integrity.
>
> 	As visualized in Figure 6, our text-stream modulation effectively injects semantic features from reference images while preserving the original structure of text-to-image outputs. This distinction allows us to maintain strong editability while achieving high fidelity, reducing the reliance on VAE features compared to approaches that alter token values directly.
>
> 2. **Using VAE-encoded image features to enhance target identity fidelity has become a widely adopted technique**
>
> 	We appreciate the reviewer's attention to our use of VAE. We agree that VAE-encoded features are widely adopted to enhance identity fidelity, but our core innovation lies not in "using VAE" but in a novel integration mechanism that addresses limitations of existing methods. Categorizing all VAE-based schemes as identical overlooks critical methodological differences.
>
> 	Recently, VAE-only methods demonstrate strong performance in specific settings, particularly excelling at controlled generation for individual subjects. However, when extended to multi-subject control scenarios, they often have difficulty preserving consistent identity fidelity across multiple subjects, frequently resulting in blurred or merged identities.
>
> 	Our XVerse solves these issues through two key innovations:
> 	- **Hybrid Injection Strategy**: Instead of global injection: the Double-stream module uses a text-modulation adapter for precise control over specific subject or high-level semantics (posture, shape). The Single-stream module receives direct VAE injection for the refinement of local details/textures. This enables implicit learning of injection regions without explicit attention masks, thereby preserving identity while protecting image structure.
> 	- **Novel Regularization Losses**: These guide the model to distinguish identity-controlled vs. text-controlled regions, achieving clearer disentangled representations and mitigating attribute blending.
>
> 	Our ablation experiments (Due to the limitation on the length, we are unable to provide the complete ablation implementation details in this response. You can refer to our reply to Reviewer 1 for more analysis) confirm this: a VAE-only model performs far worse than our complete XVerse, proving that our hybrid injection and regularization losses together enable high-fidelity, controllable generation—our core innovation.
>
> 	| | | | | | | | Single | | | | | | Multi | | |
> 	|:---:|:---:|:---:|:---:|:---:|:---:|:---:|:---:|:---:|:---:|:---:|:---:|:---:|:---:|:---:|:---:|
> 	| EXP ID | EXP Name | Init | Steps | | AVG | DPG | ID-Sim | IP-Sim | AES | | AVG | DPG | ID-Sim | IP-Sim | AES |
> 	| 1 | Text Modulation Only | Scratch | 30K | | 57.55 | 96.31 | 22.19 | 54.74 | 56.96 | | 52.73 | 91.25 | 18.45 | 47.12 | 54.14 |
> 	| 1 | Text Modulation Only | Scratch | 40K | | 58.30 | 96.61 | 24.01 | 56.31 | 56.29 | | 53.54 | 91.03 | 21.81 | 48.20 | 53.14 |
> 	| 2 | Single-Stream VAE Only | Scratch | 30K | | 60.71 | 95.50 | 26.77 | 62.66 | 57.92 | | 53.32 | 91.02 | 14.47 | 52.55 | 55.26 |
> 	| 2 | Single-Stream VAE Only | Scratch | 40K | | 59.60 | 94.97 | 22.93 | 62.94 | 57.59 | | 53.37 | 91.34 | 13.99 | 52.77 | 55.38 |
> 	| 3 | Text Modulation + VAE | Exp 1, 30K | 10K | | 62.62 | 96.28 | 39.53 | 57.54 | 57.14 | | 55.16 | 90.52 | 26.97 | 49.15 | 54.04 |
>
> 	| | | | | | Single | | | | | | Multi | | |
> 	|:---:|:---:|:---:|:---:|:---:|:---:|:---:|:---:|:---:|:---:|:---:|:---:|:---:|:---:|
> 	| EXP | Steps | | AVG | DPG | ID-Sim | IP-Sim | AES | | AVG | DPG | ID-Sim | IP-Sim | AES |
> 	| Baseline | 30K | | 57.55 | 96.31 | 22.19 | 54.74 | 56.96 | | 52.73 | 91.25 | 18.45 | 47.12 | 54.14 |
> 	| Baseline - CCL | 30K | | 55.47 | 95.89 | 15.19 | 54.11 | 56.72 | | 50.72 | 92.01 | 10.83 | 46.31 | 53.76 |
> 	| Baseline - CAL | 30K | | 53.42 | 96.31 | 7.47 | 52.43 | 57.47 | | 44.51 | 77.43 | 6.34 | 40.22 | 54.05 |
>
> 	| | | | | | Single | | | | | | Multi | | |
> 	|:---:|:---:|---|:---:|:---:|:---:|:---:|:---:|---|:---:|:---:|:---:|:---:|:---:|
> 	| EXP | Steps | | AVG | DPG | ID-Sim | IP-Sim | AES | | AVG | DPG | ID-Sim | IP-Sim | AES |
> 	| Single-stream VAE Only | 30K | | 60.71 | 95.50 | 26.77 | 62.66 | 57.92 | | 53.32 | 91.02 | 14.47 | 52.55 | 55.26 |
> 	| All Blocks VAE Only | 30K | | 58.58 | 96.62 | 19.71 | 60.34 | 57.65 | | 51.35 | 88.63 | 11.15 | 50.33 | 55.29 |
>
> 	In summary, we build on VAE but solve its existing limitations via innovative architecture and specialized losses.
>
> 3. **"Copy-and-Paste" Effects**
>
> 	When there is no text controlling semantics such as expressions and poses, the model mimics the characteristics of the given condition image. However, as shown in Figure 1, our model can still retain a certain ability to edit semantics such as pose and lighting.
>
> 	During training, we did observe the copy-paste phenomenon and took targeted measures to mitigate it: the third training phase specifically incorporated cross-image data to foster more robust appearance mapping across diverse visual conditions. As our work prioritized methodological innovations, we acknowledge that the cross-image dataset, while effective to a degree, lacks the scale and diversity needed to fully eliminate such effects (as mentioned in the Limitations part). In future work, we aim to further increase data diversity, which will help minimize the copying effect and enhance the originality of the generated images.
>
> 4. **The importance of Network Architecture or Data**
>
> 	We recognize that both model structural optimization and data updates are critical to performance improvement. Our experiments confirm that even a smaller set of high-quality cross-image data can serve as a strong foundation for enhancing model performance in test scenarios, as such diversity helps the model generate more natural and coherent images across contexts. However, the transformative leap in our model’s capabilities stems from its structural innovations: by leveraging text modulation, we preserve the aesthetic quality and structural integrity inherent to text-to-image generation, while our strategic use of regularization losses and constrained VAE feature injection positions effectively mitigates the "copy-paste" artifacts often induced by VAE. These design choices enable the model to fully capitalize on high-quality data while overcoming key limitations that would otherwise hinder generation quality.
>
> 	To further validate the distinctiveness of our model design, we conducted an ablation study comparing various methods on the same dataset. The results, presented in response 2, clearly illustrate that the VAE module and the Text-Modulation adapter serve complementary functions. This controlled experiment underscores that the architectural innovations in our model are crucial for fully leveraging the data’s potential, rather than depending solely on data quality.

---

### Official Review · Reviewer_CRkS · 2025-07-01

**Clarity:** 2
**Significance:** 3
**Originality:** 2
**Rating:** 5
**Confidence:** 4

**Summary:**

This paper describes a method, XVerse, to perform multi-subject diffusion-based image customization. Motivated by the limitation in feature injection in attention layers, as commonly done in existing works, the authors proposed two key designs as their contributions: first, the use of learned offsets applied to text-stream modulation of DiTs to control identity and semantic attributes, and second, the use of VAE-encoded features for attention injection only in single blocks of FLUX model for fine-grained details. The authors also construct a multi-entity dataset for the purpose of training the proposed model.

**Questions:**

Ln 234-235: are there visual examples that show how the “cross-image data” used in the third training phase looks like. How were they curated?

Ln 196-201: how exactly are text-image attention loss formulated?

Supp D. Is there an ablation study for the region preservation loss?

Figure 3: is it hard to tell from the figure how training data is constructed. Adding step indicators, specifying the filtered output (e.g. I assume the segmented "sidewalk" is discarded), and adding more captions may help making the figure more clear.

It appears that in Figure 1, bottom-right corner, "Light" is a controllable attribute in Xverse, whereas at ln210, it is mentioned that "non-entity terms" are excluded. How does XVerse learn to control semantic concepts like lighting and style with this exclusion?

**Ethical Concerns:**

["NO or VERY MINOR ethics concerns only"]

**Final Justification:**

Based on the other reviewers' feedback and the authors' responses, I believe this work provides a solid improvement over multi-subject image editing, and thus I recommend an accept rating, on the ground that the authors will incorporate clarifications and additional details they provided in the rebuttal process.

**Limitations:**

limitations are discussed. No discussion on societal impact.

**Quality:**

3

**Strengths And Weaknesses:**

The paper is overall well-written and well grounded to improve upon recent progress in image customization.  but a few concerns remain:

Concern 1: Comparison between XVerse and TokenVerse[23]
- Much of the proposed method section 3.2 appears highly similar to TokenVerse, in terms of adding offset to token embeddings to adjust the modulation parameters, and the design to perform this injection in the W+ space. It'd help if the authors discuss difference to this work, and, despite slight difference in customization setting, compare experiment results between the two in concept-driven image generation (where concepts come from reference images, which should be supported by the proposed pipeline).

Concern 2: Copy-and-paste effect
- Figure 5: Visual results of XVerse look relatively realistic, but display more “copy-and-paste” effects compared to baselines like DreamO, which may partially explain its relatively high performance in similarity scores. It’d be useful to show diverse samples given the same prompts to shed light on the learned target distribution.

Concern 3: clarity in the method section
- the design to inject features for fine-grained details are vaguely explained. Is this done as a feature injection into the attention layers? If so, how does it mitigate the argued weakness of leveraging attention mechanism in image customization (ln21-27)?

Concern 4: quantitative results are missing in ablation study.

Based on the above concerns, the novelty of the paper appears incremental (primarily combining the designs of TokenVerse[23] and Omnicontrol[11]) but creditable. The effectiveness of the method can be validated by the evaluation, though some additional comparisons could strengthen the claim (e.g. comparison to TokenVerse, quantitative ablation study). Some details of the method are also missing, which may be a problem for reproducibility. Overall, I consider this an incremental work that shows creditable improvement on multi-subject image editing.

---

> ### Author Rebuttal · Authors · 2025-07-31
>
> Thank you for your thorough review and insightful questions. Your feedback has been invaluable in improving the clarity and depth of our work. We have responded to each of your comments below:
>
> 1. **Distinction from TokenVerse and Comparative Analysis**
>
> 	The core difference between XVerse and TokenVerse [23] lies in tuning requirements and generalization capability:
> 	- **TokenVerse**: Relies on fine-tuning on small-scale predefined datasets to learn subject-specific modulations, requiring reference one image per concept for overfitting-based customization. This limits its applicability to unseen subjects without retraining.
> 	- **XVerse**: Uses an adapter to convert reference images and their descriptions into token-specific offsets. By converting reference images into token-specific offsets, XVerse achieves control without test-time tuning, eliminating the need for training on specific subjects. As a result, XVerse can generalize any concept from reference images after just a single training.
>
> 2. **Addressing "Copy-and-Paste" Effects with Diverse Samples**
>
> 	The evaluation in XVerseBench all uses the average results of 4 seeds, which, to some extent, can evaluate the stability of the model under the same prompt. From the visualization results, it can also be seen that compared to methods like OmniGen, Uno, DreamO that only use VAE, our model maintains better editability and stability. Subsequently, we will add some generation results corresponding to different seeds under the same prompt in the supplementary documentation to further demonstrate the model's stability.
>
> 	During training, we did observe the copy-paste phenomenon and took targeted measures to mitigate it: the third training phase specifically incorporated cross-image data to foster more robust appearance mapping across diverse visual conditions. As our work prioritized methodological innovations, we acknowledge that the cross-image dataset, while effective to a degree, lacks the scale and diversity needed to fully eliminate such effects (as mentioned in the Limitations part). In future work, we aim to further increase data diversity, which will help minimize the copying effect and enhance the originality of the generated images.
>
> 3. **Clarification on Fine-Grained Feature Injection**
>
> 	Thank you for your valuable comment. The features are directly injected into the attention of single-stream blocks. Notably, within the self-attention of double-stream modules, a deliberate design choice is made: while latents and text features cannot interact with VAE features, VAE features retain the ability to perceive latents and text features for effective feature aggregation. This mechanism prevents the degradation that would occur if VAE features were directly inserted into the single stream. We will strengthen the more detailed description of VAE injection in the revised version.
>
> 	To mitigate the argued weakness of leveraging the attention mechanism in image customization, we have adopted the following strategies:
> 	- We only perform feature injection in the single-stream blocks, while using text-modulation adapters in the double blocks for semantic control. As shown in the supplementary ablation experiments, global injection would lead to a significant decline in the naturalness of the generated images and cause obvious feature fusion in multi-condition scenarios.
> 	- Two regularization losses are introduced to constrain the application scope of the VAE. This helps the model better identify the regions where different conditions should be injected, thereby reducing confusion and fusion.
>
> 4. **More Quantitative Ablation Study**
>
> 	We conducted a series of ablation studies to dissect critical components and their impacts on the framework, ensuring all experiments shared identical configurations (e.g., hyperparameters, training schedules) unless specified otherwise, and were executed on 16 NVIDIA A800 (40GB) GPUs due to computational constraints.  Due to the limitation on the length, we are unable to provide the complete ablation implementation details in this response. You can refer to our reply to Reviewer 1 for more details on the quantitative metrics and analysis.
>
> 5. **More details of method for reproducibility**
>
> 	Thank you for your valuable feedback. In the revised version, we will enhance the supplementary document by providing more comprehensive details, including the hyperparameter settings and detailed training configurations. Additionally, we plan to opensource the model and related code in the future, which we believe will further facilitate reproducibility and foster further research in this area.
>
> 6. **Formulation for text-image attention loss**
>
> 	Thank you for your question. To more effectively constrain the region of feature injection, we performed two forward propagations in each iteration: one for text-to-image generation, and the other for controlled subject generation. The text-image attention loss in XVerse is formulated as an L2 loss with normalization over the text-image cross-attention maps of the modulated model and the reference T2I branch. Specifically, it encourages the modulated model to retain attention patterns that closely match those of the T2I branch, ensuring that semantic interactions between text and image regions remain consistent and editable.
>
> 	Mathematically, it can be expressed as:
> 	$$L_{attention}=E\left[\left\| Norm\left(A_{\theta \prime}\right)-Norm\left(A_{\theta}\right)\right\| ^{2}\right],$$
> 	where $A_{\theta'}$ denotes the text-image cross-attention maps of XVerse's modulated branch, $A_{\theta}$ denotes those of the T2I branch, and $Norm(\cdot)$ represents the normalization operation applied to the attention maps. You can also refer to Sections D and E in the supplementary document to help understand these two regularization loss functions.
>
> 7. **More details for the training data**
>
> 	For human-centric cross-image data, we collected a 100K-scale single-person multi-view dataset from proprietary in-house collections, forming up to three image pairs per subject ID. These were curated by clustering facial images and selecting diverse views (e.g., facial and full-body shots) to enhance generalization. For general objects, we leveraged the Subject200K dataset, applying DINOv2 similarity filtering to ensure cross-view consistency.
>
> 	The data curation involved generating image descriptions, detecting/segmenting subject terms using Florence2 and SAM2, and aligning conditioning images with subject terms via DINOv2 scores to form cross-image pairs. This process ensures robust appearance mapping under heterogeneous visual conditions.
>
> 8. **How does XVerse learn to control semantic concepts like lighting and style with this exclusion?**
>
> 	Thank you for your question. We excluded non-entity terms (such as "light" and "style") during data processing primarily to enhance the accuracy of detection and segmentation for entities, which is crucial for effectively learning subject-specific features. XVerse's text flow modulation mechanism enables precise control over semantic concepts such as lighting and style. This is thanks to text flow modulation operating at the semantic level of text prompts, allowing the model to naturally extend control from entity features to related semantic attributes without relying on enough explicit non-entity annotations in the training data.

---

> > ### Comment · Reviewer_CRkS · 2025-08-05
> >
> > Thank you authors for the detailed response. My concerns are addressed and I don't have additional concern as for now.

---

### Official Review · Reviewer_ATo1 · 2025-07-02

**Clarity:** 4
**Significance:** 2
**Originality:** 3
**Rating:** 5
**Confidence:** 5

**Summary:**

This paper introduces XVerse, which enables the composition and control of multiple subjects. The method uses a T-Mod Adapter to extract an offset vector ∆cross from reference images and applies it in the modulation space of DiTs. This control approach in the text space not only avoids interfering with the denoising process but also extends object manipulation to higher-level semantic attributes such as pose, lighting, and style.

**Questions:**

1. In Figure 2, what do the symbols c and x specifically represent? Additionally, could you clarify how the dual-branch architecture collaborates during training?
2. If the dataset scale were reduced by half, how would this impact the model’s performance? Have you conducted any experiments or observations to assess this sensitivity?(You don't need re-training your model for this question.)

**Ethical Concerns:**

["NO or VERY MINOR ethics concerns only"]

**Limitations:**

Please refer to Weakness.

**Quality:**

3

**Strengths And Weaknesses:**

Strengths
1. Supports fine-grained control and composition of multiple objects.
2. Demonstrates strong subject ID consistency.
3. Provides valuable exploration of the modulation mechanism in DiTs, contributing technically to the community.
4. The paper clearly explains the proposed method and is easy to read.

Weaknesses
1. According to Sec.3.3, the training appears to involve a dual-branch structure, but the Method section does not clearly describe how these two branches collaborate.
2. The paper does not disclose the computational resources required for training. Given that at least 2M-scale datasets were collected, the training cost is likely substantial.
3. The ablation study only analyzes the effect of the VAE-encoded features, and the two proposed losses, which are presented as key contributions, were not sufficiently studied or validated.

---

> ### Author Rebuttal · Authors · 2025-07-31
>
> Thank you for your thorough review and insightful questions. Your feedback is invaluable for improving the clarity and detail of our work. We address each of your points below:
>
> 1. **Collaboration of the Dual-Branch Structure During Training**
>
> 	We appreciate the opportunity to clarify the synergy between our dual-branch structure. The T2I branch $V_θ$ provides a stable generation foundation, maintaining coherence and editability. The modulation branch $V_{\theta'}$ injects specific subject features to achieve precise control. During each iteration, the input information is separately fed into the T2I branch and the modulation branch for forward propagation, and the model is optimized using the regularization loss mentioned in Section 3.3. This design achieves a balance between generation fidelity and specific subject control, enabling XVerse to realize both high-fidelity generation and accurate subject customization.
>
> 2. **Computational Resources for Training**
>
> 	We acknowledge the importance of disclosing training resources. XVerse is trained efficiently using the following setup, leveraging parameter-efficient fine-tuning to reduce costs:
> 	- **Hardware**: 64×NVIDIA A800 (40GB) GPUs.
> 	- **Training Duration**: ~7 days total across three stages (Stage 1: 70K iterations-2days, Stage 2: 150K iterations-4 days, Stage 3: 10K iterations-1 day).
>
> 3. **Ablation Studies for Regularization Losses**
>
> 	We conducted ablation experiments to evaluate the impacts of the CCL and CAL regularization losses, respectively, by comparing against a baseline model (which retains both losses). All experiments share identical configurations except for the regularization loss component. Owing to computational resource constraints, each experiment was executed on 16 NVIDIA A800 GPUs (40GB memory per GPU). As shown in the table, the baseline model achieves the highest average scores on both single-subject and multi-subject benchmarks. This result underscores the effectiveness of the CCL and CAL regularization losses—since ablating either loss leads to performance degradation.
>
> 	| | | | | | Single | | | | | | Multi | | |
> 	|:---:|:---:|:---:|:---:|:---:|:---:|:---:|:---:|:---:|:---:|:---:|:---:|:---:|:---:|
> 	| EXP | Steps | | AVG | DPG | ID-Sim | IP-Sim | AES | | AVG | DPG | ID-Sim | IP-Sim | AES |
> 	| Baseline | 30K | | 57.55 | 96.31 | 22.19 | 54.74 | 56.96 | | 52.73 | 91.25 | 18.45 | 47.12 | 54.14 |
> 	| Baseline - CCL | 30K | | 55.47 | 95.89 | 15.19 | 54.11 | 56.72 | | 50.72 | 92.01 | 10.83 | 46.31 | 53.76 |
> 	| Baseline - CAL | 30K | | 53.42 | 96.31 | 7.47 | 52.43 | 57.47 | | 44.51 | 77.43 | 6.34 | 40.22 | 54.05 |
>
> 4. **Symbols in Figure 2**
>
> 	In Figure 2, $x$ is the latent image feature, and $c$ is the text conditioning feature. The two-branch structure illustrated in Figure 2 depicts the double-stream blocks of the Flux model, which is subsequently followed by the single-stream blocks that integrate both the text feature and the latent feature. Note that Figure 2 does not include the dual-branch architecture described in Section 3.3; we will provide a more detailed explanation in the revised version.
>
> 5. **Influence of Training Dataset Size and Quality**
>
> 	Thank you for your question. We recognize that dataset scale can impact model performance. A reduced dataset scale, such as halving the current size, would likely increase the risk of overfitting to the training data, potentially undermining the model’s generalization ability to unseen categories.
>
> 	However, compared to the sheer quantity of data, the diversity and quality of the dataset are more critical. Our experiments indicate that even a smaller set of high-quality cross-image data, which covers diverse scenarios and maintains cross-view consistency, can significantly enhance the model’s performance in test scenarios. This diversity enables the model to generate more natural and coherent images across different contexts.
>
> 6. **Additional: More Expanded Ablation Studies with Numerical Results**
> 	We also conducted a series of ablation studies to dissect critical components and their impacts on the framework, ensuring all experiments shared identical configurations (e.g., hyperparameters, training schedules) unless specified otherwise, and were executed on 16 NVIDIA A800 (40GB) GPUs due to computational constraints.
>
> 	- Joint Training of Text Modulation and VAE-Encoded Features
>
> 	| | | | | | | | Single | | | | | | Multi | | |
> 	|:---:|:---:|:---:|:---:|:---:|:---:|:---:|:---:|:---:|:---:|:---:|:---:|:---:|:---:|:---:|:---:|
> 	| EXP ID | EXP Name | Init | Steps | | AVG | DPG | ID-Sim | IP-Sim | AES | | AVG | DPG | ID-Sim | IP-Sim | AES |
> 	| 1 | Text Modulation Only | Scratch | 30K | | 57.55 | 96.31 | 22.19 | 54.74 | 56.96 | | 52.73 | 91.25 | 18.45 | 47.12 | 54.14 |
> 	| 1 | Text Modulation Only | Scratch | 40K | | 58.30 | 96.61 | 24.01 | 56.31 | 56.29 | | 53.54 | 91.03 | 21.81 | 48.20 | 53.14 |
> 	| 2 | Single-Stream VAE Only | Scratch | 30K | | 60.71 | 95.50 | 26.77 | 62.66 | 57.92 | | 53.32 | 91.02 | 14.47 | 52.55 | 55.26 |
> 	| 2 | Single-Stream VAE Only | Scratch | 40K | | 59.60 | 94.97 | 22.93 | 62.94 | 57.59 | | 53.37 | 91.34 | 13.99 | 52.77 | 55.38 |
> 	| 3 | Text Modulation + VAE | Exp 1, 30K | 10K | | 62.62 | 96.28 | 39.53 | 57.54 | 57.14 | | 55.16 | 90.52 | 26.97 | 49.15 | 54.04 |
>
> 	We designed three experiments to investigate joint training of text modulation and VAE-encoded feature injection. As shown in the table, Exp. 1 (Text Modulation Only) achieves average (AVG) scores of 57.55 and 52.73 on single-subject and multi-subject benchmarks, respectively, after 20K training steps. With an additional 10K training steps, its performance marginally improves to 58.30 and 53.54, respectively. In contrast, Exp. 2 (VAE-only) attains slightly higher AVG scores of 60.71 and 53.32 after 30K training steps, yet its performance deteriorates with a further 10K training steps. This observation underscores the instability and challenges of VAE feature injection, which stem from conceptual confusion and attribute blending. Notably, Exp. 3—which employs 10K joint training steps for text modulation and VAE feature injection, using Exp. 1 (30K steps) as the initial checkpoint—exhibits performance improvements from 57.55 to 62.62 (single-subject AVG) and from 52.73 to 55.16 (multi-subject AVG). Collectively, our approach substantially alleviates prevalent challenges in VAE feature injection, thereby enabling a cleaner and more disentangled generation process.
>
> 	The instability of pure VAE feature injection (Exp. 2) is evident from its lower ID-Similarity (ID-Sim) scores. Lacking semantic guidance, the VAE-only model often generates images that fail to retain the identity information of the reference image, resulting in inconsistent and lower-quality outputs. In contrast, text modulation, demonstrated in Exp. 3, serves as a crucial “stabilizer” and “guide”. It enables the VAE feature injection in a more controlled and semantically coherent manner, reducing conceptual errors and significantly enhancing the model’s stability and editability.
>
> 	 -  VAE-encoded feature injection blocks
>
> 	| | | | | | Single | | | | | | Multi | | |
> 	|:---:|:---:|---|:---:|:---:|:---:|:---:|:---:|---|:---:|:---:|:---:|:---:|:---:|
> 	| EXP | Steps | | AVG | DPG | ID-Sim | IP-Sim | AES | | AVG | DPG | ID-Sim | IP-Sim | AES |
> 	| Single-stream VAE Only | 30K | | 60.71 | 95.50 | 26.77 | 62.66 | 57.92 | | 53.32 | 91.02 | 14.47 | 52.55 | 55.26 |
> 	| All Blocks VAE Only | 30K | | 58.58 | 96.62 | 19.71 | 60.34 | 57.65 | | 51.35 | 88.63 | 11.15 | 50.33 | 55.29 |
>
> 	We conducted an ablation study to dissect the role of VAE-encoded feature injection blocks in our framework. All experiments adopted identical experimental configurations except for the injection block designs (i.e., “Single-stream VAE Only” vs. “All Blocks VAE Only” as defined in the table). As shown in Table, injecting VAE-encoded features into the single-stream module of FLUX (Single-stream VAE Only) outperformed "All Blocks VAE Only" on both single-subject and multi-subject benchmarks. For instance, in single-subject AVG, it achieved 60.71 vs. 58.58, and in multi-subject AVG, 53.32 vs. 51.35. This highlights the superiority of targeted single-stream VAE feature injection.

---

### Official Review · Reviewer_S48T · 2025-07-17

**Clarity:** 2
**Significance:** 2
**Originality:** 3
**Rating:** 4
**Confidence:** 3

**Summary:**

The paper proposes a solution to multiple subject control in DiT based image synthesizer and a benchmark dataset for evaluating single & multiple subject control tasks. The method achieve high-fidelity control of multiple (6+) subjects and their semantic attributes (poses, highlights and styles) during image generation.

**Questions:**

What about the stableness and performance under multiple subject injection? Till how many of subjects will the method still work in a stable way? It is better to provide evaluations under different subject amount.

If the authors can clear some of my doubts (about novelty and ablation studies), I would be happy to raise my scores.

**Ethical Concerns:**

["NO or VERY MINOR ethics concerns only"]

**Final Justification:**

After the rebuttal, the authors completed the most of missing experiments on ablations I required, which proves the integrity of the paper. Also, some of my doubts and misunderstandings are cleared. I have raised my rating for the paper.

**Limitations:**

The method is architecture-specific to the DiT Models.

**Paper Formatting Concerns:**

None.

**Quality:**

3

**Strengths And Weaknesses:**

Strengths:

The method achieves SOTA on both single and multiple subject control tasks (numerically and visually).

The method is designed in a dedicated way for preventing different subjects interfering with each others, and is stable across various scenes.

The method proposes to convert each reference images into subject-specific modulation offsets, which is a novel idea for token-level modulation.

Weaknesses:

The writing of this paper is a bit confusing, the authors spends a lot of space to introduce the well-known AdaLayerNorm module in DiT, instead of introducing the module Perceiver Resampler used by the method's key mechanism. Also, some details is not well-explained in the paper, e.g. "only injecting vae injection into a single block". So which block? Why this block? Why not two or more blocks? Any further ablation studies?

The ablation study is not enough (maybe due to the complex training pipeline), no numerical results indicate the performance loss on each mechanism (like the effectiveness of global-local text-offsets).

As for the novelty, the main contribution is the injection of token-level modulation offsets, while the other of key mechanism have precedents, which lowers the overall novelty of the paper.

---

> ### Author Rebuttal · Authors · 2025-07-31
>
> Thank you for your insightful comments and valuable suggestions, which have helped us better refine our work. We address each point as follows:
>
> 1. **Clarification on the Focus of AdaLayerNorm and Perceiver Resampler**
>
> 	Thank you for your valuable comment. We devoted significant space to AdaLayerNorm because our work extends its application beyond its original purpose in DiT. Since AdaLN is known for accelerating model convergence [15], many works have not considered that it can be used for subject control. In contrast, we leverage it in XVerse to achieve fine-grained control over subject identity without extra test-time tuning, which is a key innovation of our method.
>
> 	In contrast, the Perceiver Resampler is a more established framework and not our core contribution. However, we acknowledge the need for more detail. In the revised version, we will elaborate on how the Perceiver Resampler is utilized within our text-stream modulation adapter to enhance clarity.
>
> 2. **Details on VAE Feature Injection into a Single Block**
>
> 	Thank you for your feedback. The FLUX architecture includes two structures:
> 	- The double-stream module, consisting of 19 Transformer blocks, processes text and image streams independently, as shown on the right side of Figure 2. Each stream is equipped with its own projection layers, modulation parameters, and normalization layers.
> 	- The single-stream module, deployed after the dual-stream module, contains 38 Transformer blocks. It unifies text and image features without strict modality distinction.
>
> 	In XVerse, the VAE encoded features do not interact with the latent features and text features in the double-stream module and are injected exclusively into the single-stream module of FLUX. This choice mitigates VAE's potential disruption to attention mechanisms and text-to-image aesthetics/structure. We conducted ablation studies on block selection (as shown below), confirming that limiting VAE injection to single-stream blocks best preserves generation quality, editability, and generalization. We will add more details to the revision.
>
> 3. **Expanded Ablation Studies with Numerical Results**
>
> 	We conducted a series of ablation studies to dissect critical components and their impacts on the framework, ensuring all experiments shared identical configurations (e.g., hyperparameters, training schedules) unless specified otherwise, and were executed on 16 NVIDIA A800 (40GB) GPUs due to computational constraints.
>
> 	 - VAE-encoded feature injection blocks
>
> 	| | | | | | Single | | | | | | Multi | | |
> 	|:---:|:---:|---|:---:|:---:|:---:|:---:|:---:|---|:---:|:---:|:---:|:---:|:---:|
> 	| EXP | Steps | | AVG | DPG | ID-Sim | IP-Sim | AES | | AVG | DPG | ID-Sim | IP-Sim | AES |
> 	| Single-stream VAE Only | 30K | | 60.71 | 95.50 | 26.77 | 62.66 | 57.92 | | 53.32 | 91.02 | 14.47 | 52.55 | 55.26 |
> 	| All Blocks VAE Only | 30K | | 58.58 | 96.62 | 19.71 | 60.34 | 57.65 | | 51.35 | 88.63 | 11.15 | 50.33 | 55.29 |
>
> 	We conducted an ablation study to dissect the role of VAE-encoded feature injection blocks in our framework. All experiments adopted identical experimental configurations except for the injection block designs (i.e., “Single-stream VAE Only” vs. “All Blocks VAE Only” as defined in the table). As shown in Table 1, injecting VAE-encoded features into the single-stream module of FLUX outperformed "All Blocks VAE Only" on both single-subject and multi-subject benchmarks. For instance, in single-subject AVG, it achieved 60.71 vs. 58.58, and in multi-subject AVG, 53.32 vs. 51.35. This highlights the superiority of targeted single-stream VAE feature injection.
>
> 	 - Joint Training of Text Modulation and VAE-Encoded Features
>
> 	| | | | | | | | Single | | | | | | Multi | | |
> 	|:---:|:---:|:---:|:---:|:---:|:---:|:---:|:---:|:---:|:---:|:---:|:---:|:---:|:---:|:---:|:---:|
> 	| EXP ID | EXP Name | Init | Steps | | AVG | DPG | ID-Sim | IP-Sim | AES | | AVG | DPG | ID-Sim | IP-Sim | AES |
> 	| 1 | Text Modulation Only | Scratch | 30K | | 57.55 | 96.31 | 22.19 | 54.74 | 56.96 | | 52.73 | 91.25 | 18.45 | 47.12 | 54.14 |
> 	| 1 | Text Modulation Only | Scratch | 40K | | 58.30 | 96.61 | 24.01 | 56.31 | 56.29 | | 53.54 | 91.03 | 21.81 | 48.20 | 53.14 |
> 	| 2 | Single-Stream VAE Only | Scratch | 30K | | 60.71 | 95.50 | 26.77 | 62.66 | 57.92 | | 53.32 | 91.02 | 14.47 | 52.55 | 55.26 |
> 	| 2 | Single-Stream VAE Only | Scratch | 40K | | 59.60 | 94.97 | 22.93 | 62.94 | 57.59 | | 53.37 | 91.34 | 13.99 | 52.77 | 55.38 |
> 	| 3 | Text Modulation + VAE | Exp 1, 30K | 10K | | 62.62 | 96.28 | 39.53 | 57.54 | 57.14 | | 55.16 | 90.52 | 26.97 | 49.15 | 54.04 |
>
> 	We designed three experiments to investigate joint training of text modulation and VAE-encoded feature injection. As shown in the table, Text Modulation Only model achieves average (AVG) scores of 57.55 and 52.73 on single-subject and multi-subject benchmarks, respectively, after 30K training steps. With an additional 10K training steps, its performance marginally improves to 58.30 and 53.54, respectively. In contrast, VAE-only attains slightly higher AVG scores of 60.71 and 53.32 after 30K training steps, yet its performance deteriorates with a further 10K training steps. This observation underscores the instability and challenges of VAE feature injection, which stem from conceptual confusion and attribute blending. Notably, Exp. 3—which employs 10K joint training steps for text modulation and VAE feature injection, using Exp. 1 (30K steps) as the initial checkpoint—exhibits performance improvements from 57.55 to 62.62 (single-subject) and from 52.73 to 55.16 (multi-subject). Collectively, our approach substantially alleviates prevalent challenges in VAE feature injection, thereby enabling a cleaner and more disentangled generation process.
>
> 	The instability of pure VAE feature injection is evident from its lower ID-Similarity scores. Lacking semantic guidance, the VAE-only model often generates images that fail to retain the identity information of the reference image, resulting in inconsistent and lower-quality outputs. In contrast, text modulation, demonstrated in Exp. 3, serves as a crucial “stabilizer” and “guide”. It enables the VAE feature injection in a more controlled and semantically coherent manner, reducing conceptual errors and significantly enhancing the model’s stability and editability.
>
> 	 - Impacts of CCL and CAL Regularization Losses
>
> 	| | | | | | Single | | | | | | Multi | | |
> 	|:---:|:---:|:---:|:---:|:---:|:---:|:---:|:---:|:---:|:---:|:---:|:---:|:---:|:---:|
> 	| EXP | Steps | | AVG | DPG | ID-Sim | IP-Sim | AES | | AVG | DPG | ID-Sim | IP-Sim | AES |
> 	| Baseline | 30K | | 57.55 | 96.31 | 22.19 | 54.74 | 56.96 | | 52.73 | 91.25 | 18.45 | 47.12 | 54.14 |
> 	| Baseline - CCL | 30K | | 55.47 | 95.89 | 15.19 | 54.11 | 56.72 | | 50.72 | 92.01 | 10.83 | 46.31 | 53.76 |
> 	| Baseline - CAL | 30K | | 53.42 | 96.31 | 7.47 | 52.43 | 57.47 | | 44.51 | 77.43 | 6.34 | 40.22 | 54.05 |
>
> 	We evaluated CCL and CAL regularization losses by comparing against a baseline (retaining both losses). As shown in the table, the baseline model achieves the highest average scores on both single-subject and multi-subject benchmarks. This result underscores the effectiveness of the CCL and CAL regularization losses—since ablating either loss leads to performance degradation.
>
> 4. **Novelty: Distinction from Prior VAE-Based Methods**
>
> 	Thank you for your insightful feedback regarding the novelty of our work. We agree that VAE-encoded features are widely adopted to enhance identity fidelity, but our core innovation lies not in "using VAE" but in a novel integration mechanism that addresses limitations of existing methods. When extended to multi-subject control scenarios, traditional VAE-only methods often have difficulty preserving consistent identity fidelity across multiple subjects, frequently resulting in blurred or merged identities.
>
> 	Our XVerse solves these issues through two key innovations:
> 	- **Hybrid Injection Strategy**: Instead of global injection: the Double-stream module uses a text-modulation adapter for precise control over specific subject or high-level semantics (posture, shape). The Single-stream module receives direct VAE injection for the refinement of local details/textures. This enables implicit learning of injection regions without explicit attention masks, thereby preserving identity while protecting image structure.
> 	- **Novel Regularization Losses**: These guide the model to distinguish identity-controlled vs. text-controlled regions, achieving clearer disentangled representations and mitigating attribute blending.
>
> 	Our ablation experiments (as shown in Response 3) confirm this: a VAE-only model performs far worse than our complete XVerse, proving that our hybrid injection and regularization losses together enable such a high-fidelity, controllable generation. In summary, we build on VAE but solve its existing limitations via innovative architecture and specialized losses.
>
> 5. **Stability Under Multi-subject Injection**
>
> 	Thanks for your suggestion. We evaluated stability across 1Instance, 2-Instance and 3-Instance scenarios in XVerseBench. The results are presented as the average scores for editability (DPG-score), ID/IP similarity, and aesthetic quality:
>
> 	| Method | 1-Instance | 2-Instance | 3-Instance |
> 	|:---:|:---:|:---:|:---:|
> 	| OmniGen | 67.00 | 60.94 | 51.91 |
> 	| UNO | 64.65 | 58.63 | 52.11 |
> 	| DreamO | 69.44 | 64.45 | 50.97 |
> 	| XVerse | 71.13 | 64.85 | 62.96 |
>
> 	The table illustrates that the model's performance varies with the number of subjects. Nevertheless, XVerse consistently outperforms state-of-the-art methods across all three protocols and demonstrates significantly greater stability, exhibiting an average performance decline of less than 10% even when applied to up to three subjects. You can also check out the examples in Figure 1. These show that our model can still produce high-quality images even when it is used with multiple subjects.

---

> > ### Comment · Reviewer_S48T · 2025-08-06
> >
> > Thank for the authors' responsive feedback, which clears most of my doubts and completes the data required. This is a reasonable work.

---

### Decision · Program_Chairs · 2025-09-17

**Decision:**

Accept (poster)

**Comment:**

This paper receives positive ratings of (4, 4, 5, 5). The reviewers generally acknowledge the performance of the proposed method, but raised on questions on ablations, designs, and writing. After reading the paper, review and rebuttal, the AC finds no reason to overturn the decision of reviewers. An acceptance is recommended. The authors are advised to revise the paper according to the reviewers' comment.